# An Updated Review of the Marine Ornamental Fish Trade in the European Union

**DOI:** 10.3390/ani14121761

**Published:** 2024-06-11

**Authors:** Monica Virginia Biondo, Rainer Patrick Burki, Francisco Aguayo, Ricardo Calado

**Affiliations:** 1Fondation Franz Weber, 3011 Bern, Switzerland; 2asdfg IT, Fluh 86, 3204 Rosshaeusern, Switzerland; 3Faculty of Higher Studies Cuautitlán, National Autonomous University of Mexico, Mexico City 54714, Mexico; 4ECOMARE—Laboratory for Innovation and Sustainability of Marine Biological Resources, CESAM—Centre for Environmental and Marine Studies, Department of Biology, Santiago University Campus, University of Aveiro, 3810-193 Aveiro, Portugal

**Keywords:** coral reef fish, international aquarium trade, management, value, TRAde Control and Expert System TRACES

## Abstract

**Simple Summary:**

Marine aquarium keeping is a popular hobby that fuels a global industry that still heavily relies on the sourcing of wild organisms, mostly from tropical coral reefs. The European Union (EU) is one of the largest import markets for marine ornamental fish. Despite being mandatory and already fully digital, the record-keeping of what species are imported in what numbers from which exporting countries remains blurry. The present work presents curated and consolidated data reporting the value, the exporting and importing countries, and the number of specimens, species, and families of marine ornamental fish imported to the EU between 2014 and 2021. A 24-million-euro annual trade value was recorded, and 26 million specimens were imported from more than 60 countries (mostly Indonesia, the Philippines, and Sri Lanka). A watchlist is presented to provide guidance to stakeholders on which marine ornamental fish species being traded most likely require closer monitoring due to their potential impact through trade. The European TRAde Control and Expert System (TRACES) requires some minor tunning to enable authorities to easily monitor the imports of marine ornamental fish into the EU, thus allowing an unprecedented insight into this and other trade activities targeting wildlife.

**Abstract:**

Wild-caught fish from coral reefs, one of the most threatened ecosystems on the planet, continue to supply the marine aquarium trade. Despite customs and veterinary checks during imports, comprehensive data on this global industry remain scarce. This study provides consolidated data on the largest import market by value, the European Union (EU): a 24-million-euro annual trade value, detailing the main exporting and importing countries, as well as the species and families of the 26 million specimens imported between 2014 and 2021. A watchlist alert system based on the number of specimens traded, import trends, and vulnerability index according to FishBase and the IUCN Red List conservation status is presented, providing key information on which species should require closer scrutiny by authorities. While the European TRAde Control and Expert System (TRACES) electronically monitors the movement of live animals to respond quickly to biosecurity risks, one-third of marine ornamental fish imported lack species-level information. With minor adjustments, TRACES holds the potential to significantly enhance data granularity and the monitoring of wildlife trade, with marine ornamental fish being an interesting case study to validate this approach.

## 1. Introduction

The European Union (EU) is a major player in the global market for wild marine ornamental fish, both in terms of value and number of specimens. The EU Parliament’s resolution of 5 October 2022 emphasized the importance of addressing the trade of marine ornamental fish and its monitoring, particularly as most marine ornamental fish are wild-caught [1]. Additionally, during the 18th Conference of the Parties to the Convention on International Trade of Endangered Species of Wild Fauna and Flora (CITES) in August 2019, it was recognized that the international trade of marine ornamental fish required closer examination due to its large scale, lack of regulation, and inadequate monitoring [2]. A workshop involving CITES parties, industry representatives, experts, and NGOs only took place from 7th to 10th May 2024 in Brisbane, Australia, due to the COVID-19 pandemic; its conclusions will be discussed at the CITES Animals Committee meeting in July 2024 and the decisions taken at the 20th CITES conference in 2025 [3].

Since 2004, the EU has electronically monitored the movement of live animals, plants, and food from third-party countries (non-EU countries) to the EU using the TRAde Control and Expert System (TRACES). All fish imported into the EU are transported by air freight, primarily from southeast Asia (e.g., Indonesia, the Philippines, and Sri Lanka) [4,5]. Upon arrival in the EU, these specimens undergo customs clearance, as well as veterinary inspection, and must be registered in TRACES before being collected by wholesalers or buyers [6]. TRACES is an online platform used for sanitary certification and traceability of imports of live animals, animal products, food, and feed into Europe. It helps mitigate biosecurity risks and disease outbreaks. This system enables the EU to respond quickly to biosecurity risks like potential threats posed by zoonosis and invasive species [7]. In 2014, the monitoring of marine ornamental fish through TRACES became possible with the introduction of the “Harmonized System code 03011900 for Live ornamental fish (excluding freshwater)”, which aggregates marine ornamental fish upon import and requires data at the species level. However, since 2019, TRACES has also allowed information to be input at the genus levels, as well as family and even order. These additional reporting options make it harder to assess biosecurity risks associated with marine ornamental fish imports, and EU authorities have already been advised to correct this caveat [5,6]. Despite these shortcomings, the TRACES data collection system provides an unparalleled opportunity to gain an in-depth overview of the marine ornamental fish trade in the EU.

The global trade of marine ornamental fish, which has been ongoing for nearly a century, has never been effectively monitored [8,9,10,11,12,13]. Previous attempts to increase transparency and oversight within the industry have been unsuccessful [14,15,16]. This complex industry involves over 60 exporting countries and targets more than 2000 species [12], but the environmental impacts of harvesting millions of coral reef fish per year for this trade have largely been overlooked [17,18,19,20]. Some species have already been negatively impacted by the marine aquarium trade [21]. For example, the Banggai cardinalfish (*Pterapogon kauderni*) was discovered by the aquarium industry in the 1990s and has since then become perilously close to extinction due to its high demand by this industry [20]. In consequence, it was listed as “endangered” on the IUCN Red List in 2007, and the United States of America (USA) placed it on its Endangered Species Act and listed it as “threatened” in 2016 [22]. Presently, the USA is calling for additional protection by proposing to ban imports and exports of wild and captive-bred specimens of *P. kauderni* [23]. Another documented example of the impact of this trade on some of the species it targets is the overfishing of bluestreak cleaner wrasse (*Labroides dimidiatus*) and its fostering of biodiversity loss on coral reefs [17,24,25].

Ongoing projected climate change impacts coral reefs and the marine ornamental fish species they host [26,27,28,29,30,31,32]. The impacts of destructive fishing practices used in the marine aquarium industry, such as cyanide fishing, are further exacerbated by warmer waters, as suggested by some laboratory studies [33]. High mortality rates along the supply chain also contribute to the decline of marine ornamental fish populations, as extra specimens need to be collected to cover such losses [10,13,20,34,35,36,37].

It is, therefore, important to make updated data available to the scientific community and decision-makers as soon as possible so decisions on the trade of marine ornamental fish can be made using the best scientific evidence available to date, supported by fact and not emotive and alarmist opinion. At present, most studies working on this topic are using trade data that are more than 10 years old and mostly from the USA market [38,39], which may blur the global patterns of this activity. In this study, we analyzed data from the statistical office of the European Union, EuroStat, and the United Nations Commodity Trade Statistics Database (UN Comtrade) data to assess the monetary value of the global trade in marine ornamental fish, the importance of the EU demand in this trade, and the key trends of this market, which predominantly relies on wild-caught specimens. Additionally, we utilized TRACES data from 2014 to 2017 [5] and analyzed data from 2018 to 2021 (as 2021 is presently the last year available with revised and consolidated data). We retrieved information available on country of origin and destination (i.e., export and import country), species diversity, number of specimens, and trade trends. By considering the species’ vulnerability according to FishBase and their conservation status according to the IUCN Red List, we developed an alert system, a three-parameter list called Watchlist, which includes a previous study covering data from 2014 to 2017 [5]. This Watchlist was then extended to a WatchlistPLUS (which includes a linear regression for estimating the time trend in a number of specimens traded) that allows the ranking species that may be at risk of overexploitation due to the global marine ornamental fish trade’s impact driven by EU imports.

## 2. Materials and Methods

### 2.1. Data on Marine Ornamental Fish Value

To assess the economic value of trade in marine ornamental fish, we first examined the import values at the global and regional levels using the nominal value of imports (in USD) from the United Nations Commodity Trade Statistics Database (UN Comtrade database). This source is compiled by the United Nations Statistics Division from detailed global annual and monthly trade statistics by product and trading partner, covering approximately 195 countries and representing more than 99% of the world’s merchandise trade [40]. We extracted import values for the product classification “Harmonized System code 03011900 Live ornamental fish (excluding freshwater)” (HS 03011900) for the period 2014–2021. Global imports reported by country were aggregated by region based on the World Bank’s regional classification [41]. The People’s Republic of China was separated from the rest of East Asia and the Pacific Islands region because of its size and for showing a distinctly different (i.e., rapidly increasing) trend with respect to the Asia-Pacific region as a whole. For the EU region, we included the 27 EU countries and the United Kingdom (UK) (up to the end of 2020, as beyond this date, the UK was no longer an EU member-state), as well as Iceland, Norway, San Marino, and Switzerland. We converted import values into EUR using the annual average exchange rate of the USD.

It must be noted that import values from UN Comtrade have known biases, there is considerable under-reporting, and import values are most likely over-estimated, as many products are re-exported [42,43]. Correcting this bias directly was not possible since re-exports are practically unreported in this database. This is an important source of bias given the high level of trade within the EU, especially as imports from the rest of the world that are re-exported to another EU member state. As described in detail by Leal et al. [4] (2015), the structure of imports within the EU reflects a high degree of specialization in the trade of marine ornamental fish and allows to clearly distinguish between exporters (e.g., the UK, Netherlands, Germany) and importer countries (e.g., Spain, Italy). To avoid re-exports from being double counted as imports, we approximated the value of extra-EU imports by taking the percentage they represent in total EU imports from the EuroStat database (as reported in the Statistical Office of the European Communities’ database EU trade since 1988 by HS2-4-6 and CN8 (ds-045409), EuroStat 2023) [44]. We estimated the value of extra-EU imports by applying this percentage to the value of total EU imports as reported in UN Comtrade (Converted into EUR, total import values obtained from UN Comtrade were −5.9% lower (on average, 2014–2021) than those reported by the EuroStat for the same product classification. Part of the variation is due to movements in the exchange rate). 

Import values are only a fraction of the market value. After being imported, value is added to the product (in this case, marine ornamental fish) as wholesalers and retail traders add their costs to the selling price. This added value can be approximated in turn by calculating the net exports and the difference between EU exports and imports as reported in the UN Comtrade. The trade value chain ends with final consumers, and this final demand is a better approximation of the size of the market. To approximate the value of the final demand for ornamental marine fish in the EU, we took the value of extra-EU imports as described above and added the value of net exports. The latter can be interpreted as a measure of the costs added by traders to the final price. To calculate average prices in Euros, we used the extra-EU import value and the total import value from EuroStat. All figures in EUR were discounted for inflation using the Harmonized Index of Consumer Prices (EuroStat, 2023 dataset: HICP—annual data, average index, and rate of change) set to 2020 = 100 so that all values were comparable over time.

### 2.2. Data on Marine Ornamental Fish Species and Numbers

As for data from 2014 to 2017 [5], the EU Directorate-General for Health and Food Safety (DG SANTE) provided Excel files containing data on marine ornamental fish imported to Europe from 2018 to 2021, which were obtained from the European TRAde Control and Expert System TRACES using “HS 03011900 Live ornamental fish (excluding freshwater)”. Introduced in 2004, TRACES is widely utilized in approximately 90 countries by 113,000 users (government agencies, exporting and importing businesses, and official veterinarians) and is available in 39 languages [7]. It facilitates cooperation between EU and non-EU authorities. Since 2014, TRACES has been gathering data specifically for marine ornamental fish under the category “HS 03011900 Live ornamental fish (excluding freshwater)”, which was previously grouped more broadly as “otra pesca” [8].

Traders are required to be registered with TRACES and complete customs documents, which also physically accompany consignments. The import goods are declared at the border of any EU member state, as well as Switzerland, Norway, Iceland, or San Marino, by entering the freight details in the web interface of TRACES. TRACES is the only tool for collecting information on the number of specimens and diversity of marine ornamental fish in the EU region, although it does not secure true traceability for these marine organisms.

TRACES documents are web-based and must be completed online. Until the end of 2019, customs documents were titled “Common Veterinary Entry Document Animals” (CVEDA) and then became the harmonized “Common Health Entry Document Animals” (CHEDA), designed specifically to carry out health checks at borders. This approach led to an adaptation of TRACES data being collected for marine ornamental fish with regards to data for 2014–2017, allowing the import of Excel files and making possible the use of taxonomic higher-level data, including at the order level. TRACES records the species of marine ornamental fish in a predetermined pull-down list field called “species”, which may either contain the full scientific name or only a genus, family, or order name. This possibility decreases data granularity, making analysis of all species traded a more challenging task. Since 2019, xls or cvs files that contain species-level information can be directly imported from TRACES, an important feature that facilitates data mining and analysis. Also, widely used common names have been changed to scientific names.

The data provided by the EU contained freshwater ornamental fish, invertebrates, and other non-fish. Species from land-locked countries or countries with no tropical waters were retained, as these countries could represent commercial transit hubs for the ornamental trade. The record signaling 80,000 of the Mediterranean moray eels (*Muraena helena*) shipped from Israel to Denmark were removed from the watchlists as these were not clearly destined for aquaria; as *M. helena* is a regular food fish and because of its size, only a few are kept in public aquariums. One record of a shipment destined for the UK but having the USA as its destination country was changed to the UK, as the border inspection post was in the UK. Countries with multiple possible names were harmonized: “United Kingdom (Northern Ireland)” and “The Netherlands” were changed to “UK” and “Netherlands”, respectively.

All scientific fish names were checked by using the World Register of Marine Species (WoRMS; http://www.marinespecies.org (accessed on 23 September 2023)) and FishBase (http://www.fishbase.org (accessed on 23 September 2023)), the two main global species databases of fish species that are kept up to date. Records of fish lacking their complete genus or species identification were allocated to their family, as well as all fish with complete species names, by using FishBase and WoRMS.

TRACES data were cleaned by using information from FishBase by filtering out fish that did not match “saltwater” AND (“tropical” OR “subtropical”) AND “reef-associated” plus “saltwater” AND (“tropical” OR “subtropical”) AND “demersal” AND “aquarium”. Chichlids (Cichlidae) or archer fish (Toxotidae) are primarily freshwater fish, but FishBase either places the species as “reef-associated” (brackish) and/or as migratory ocean-river. Flatfish (Scophthalmidae) are commonly traded as food fish, but some specimens were also imported as ornamentals and were, therefore, considered as well in the present study. Information on origin and destination, number of specimens traded, and species diversity were analyzed. To describe exporting countries, species, and family diversity, we used the Shannon–Wiener Index (H’), as well as the Shannon Evenness Index (E’). While the first allows us to determine how diverse the range of exporting countries, species, and families of marine ornamental fish are on the imports being addressed, the second allows us to determine their relative abundance and infer if some countries, species, or families of marine ornamental fish being imported dominate these records.

### 2.3. Trends in Number of Specimens Traded and Watchlists

Productivity–susceptibility analysis (PSA) has been employed to assess the vulnerability of wild-captured marine ornamental fish [39,45,46,47] to identify species on the IUCN Red List likely threatened by international trade [48] or calculating relative exploitation ‘risks’ exclusively using trade data [49]. However, the present work builds upon a previously assembled and validated dataset, and the authors aimed to compare the most recent data retrieved from TRACES that have already been consolidated (2018–2021) to older data already curated by the authors (2014–2017) [5]. As such, the authors decided to use the same methods detailed in Biondo and Burki 2019 [5] to produce a Watchlist, but we have also refined this approach by using linear regression of the eight years of data being considered, resulting in WatchlistPLUS.

For the Watchlist, the score for each species was evaluated using three parameters, the number of specimens imported per year, vulnerability according to FishBase, and the IUCN Red List conservation status [5], as both databases provide complementary information on fish vulnerability [50] and, despite potential shortcomings [51,52], these are the most reliable and consolidated information available for fish species. The median number of specimens traded was normalized, assigning a value of 100 to the species with the highest eight-year median trade volume. The IUCN Red List categories were converted into numerical values as follows: “least concern” (LC) = 0, “near threatened” (NT) = 20, “vulnerable” (VU) = 40, “endangered” (EN) = 60, “critically endangered” (CR) = 80, and “extinct in the wild” (EW) = 100. “Extinct” (EX) was not assigned a value since trading an extinct species is not possible. For “data deficient” (DD) or “not evaluated” (NE) species, the IUCN preamble states that “until such time as an assessment is made, taxa listed in these categories should not be treated as if they were non-threatened. It may be appropriate (especially for “data deficient” forms) to give those species the same degree of attention as threatened taxa, at least until their status can be assessed”. For this reason, as the habitat of marine ornamental fish and coral reefs is threatened [31,32], these categories were handled as “vulnerable” (VU) and received the numerical value of 40. FishBase computes a vulnerability score for each species, representing its resilience to external factors and is calculated using selected life-history parameters, with a score ranging from 0 to 100. A score was determined for each species by summing the normalized parameter values. The higher the score, the more potentially exposed the species could be to overexploitation through the fishing effort to supply the marine aquarium trade. As every category is normalized to a value between 0 and 100, the theoretical maximum would be a score of 300. By 2022, the IUCN Red List had re-evaluated 449 species, with the following results: one species was newly rated as “endangered” (EN), four as “vulnerable” (VU), two as “near threatened” (NT), 426 as “least concern” (LC), and 16 as “data deficient” (DD). This information was updated in the data spreadsheet prior to analysis.

The WatchlistPLUS was produced using the same three parameters detailed above for the Watchlist but included a linear estimate of the trend in the number of specimens traded over the eight years (slope). The linear regression was tested for significance with a weighted R-squared for variance explanatory power and a *t*-test on the coefficient significance. Only species with a *p*-value for the t-statistic of less or equal to 0.05 were retained. The time series has only eight observations, which may be just enough to ensure statistical significance if the variance is low [53]. The WatchlistPLUS revealed that several species of marine ornamental fish slope estimations were not meaningful due to high variance or missing data. This method will, however, improve in accuracy as the database spans over time.

## 3. Results

### 3.1. Import Values of Marine Ornamental Fish

According to UN Comtrade, between 2014 and 2021, the EU total import values of marine ornamental fish reached EUR 24 million on average per year (at 2020 constant prices). This value includes the UK (up to the end of 2020), as well as Iceland, Norway, San Marino, and Switzerland (for the whole period 2014–2021). If we consider imports coming strictly from outside the EU-27 (“extra-EU”, Figure 1), the figure is reduced to EUR 12.1 million on average between 2014 and 2021. Although its share in the global imports of these organisms has diminished (from 40.5% in 2014), the EU still accounts for 35.4% of the global imports in 2021 (Figure 2). As shown in Figure 3, EU import values have a cyclical trajectory with an almost constant trend, while trade in the rest of the world exhibits a clear increasing trend, driven mostly by the USA, Canada, and Chinese Markets. The value of extra-EU imports decreased by 26.8% between 2014 and 2021 (Figure 4), which is aligned with the 59.9% decrease in the number of specimens reported (Figure 5). Final demand, however, increased by 14.6% during the same period, averaging EUR 35.4 million in 2014–2021 (Figure 4), and did not exhibit a reducing trend during the whole period of 2014–2021. Based on the observed data for value and the number of specimens, average import prices doubled from EUR 2.1 to EUR 4.6 per specimen between 2014 and 2021 when considering only extra-EU imports but more than doubled from EUR 6.9 to EUR 19.8 per specimen when final demand was considered (Figure 6).

### 3.2. Country of Origin, Destination, and Specimens

From 2014 to 2021, a total of 61 countries exported marine ornamental fish to the EU. The main exporting country was Indonesia, which exported 33.5% of its shipments and an average of 1,394,208 specimens exported per year (Table 1). With regards to the number of specimens exported, Indonesia was followed by the Philippines, with an average of 529,076 fish and Sri Lanka, with 266,945 fish; however, in terms of shipments, Sri Lanka displayed a higher number than the Philippines (16.2% and 12.2%, respectively). The same trend was recorded for the Maldives, Singapore, and Israel, with the Maldives exporting higher fish numbers but Singapore and Israel making more shipments. 

The three main exporting countries, Indonesia, the Philippines, and Sri Lanka, accounted for 61.9% of shipments and 68.7% of specimens and, together with the USA, Kenya, the Maldives, Singapore, and Israel, the combined total was 91.7% of all shipments and 91.6% of all specimens of marine ornamental fish entering the EU (Table 1). These eight countries represented 83.9% of import value (Table 1). In terms of the value of trade, the USA came second and the Philippines third in importance after Indonesia, shipping fewer specimens but fetching higher average prices. The H’ values concerning the diversity of exporting countries remained stable over 2014–2020 at 1.96, although in 2021, a lower value was recorded (1.7) (Appendix A). A similar trend was also recorded for E’ values, which averaged 0.48 for 2014–2020 and dropped to 0.42 in 2021 (Appendix A).

With regards to value, on average, Indonesia shipped 765 specimens per shipment at a value of EUR 2.6 per fish, whereas the Philippines shipped more species per shipment, 797 specimens, with each fish costing EUR 1.7 (Table 2). The most expensive fish were shipped from the USA at an average value of EUR 5.6 per fish (Table 2).

In total, 43,582 shipments with 25,503,345 specimens were imported into Europe, with an average of 3,187,918 specimens a year (Figure 5). The annual number of imported specimens decreased by 59.9% between 2014 and 2021 (Figure 5 and Figure 6). Although the number of specimens from Indonesia decreased between 2014 and 2021, their overall value increased (Figure 7). For Kenya, the number of specimens exported increased but not in the same proportion as their value (Figure 7). Thirty European countries imported marine ornamental fish between 2014 and 2021, including Iceland, Norway, San Marino, and Switzerland, which are not part of the EU, along with the UK, which left the EU at the end of 2020 (Table 3).

The EU country importing most marine ornamental fish was the UK (6,480,759 specimens in total), except for 2021; it was followed by the Netherlands (5,106,963 specimens), which in 2021 imported more marine ornamental fish than the UK. The Netherlands was followed by Germany, with 3,942,112 specimens. These three countries accounted for 60.9% of all imports of marine ornamental fish into the EU between 2014 and 2021 (Table 3.). With the inclusion of Italy and France, these five countries alone imported a total of 80.5% of all marine ornamental fish imported into the EU (Figure 8, Table 3).

### 3.3. Diversity and Richness of Imported Marine Ornamental Fish

Between 2014 and 2021, fish species from 120 families were imported into Europe, ranging from 68 families in 2014 to 64 families in 2017 (the lowest number recorded) and peaking at 90 families in 2021. The top 12 families recorded accounted for 92.4% of all traded marine ornamental fish imported into the EU in terms of number of specimens (Figure 9). The H’ values concerning the diversity of marine ornamental fish families imported into EU countries remained stable over 2014–2021 (averaging at 2.29), with a similar trend being recorded for E’ values (averaging 0.48 for the same period). Nonetheless, it is worth mentioning that one of the lowest values recorded (0.46) was in 2021 (Appendix A). The family Labridae featured the highest number of imported species (210), followed by Pomacentridae (142), which was also the most traded family in number of specimens (Figure 9).

Between 2014 and 2021, 1452 species of marine ornamental fish were imported into the EU. However, of the 25,503,345 specimens imported, only 17,770,326 specimens (69.7%) were registered at the species level in the TRACES database.

The blue-green damselfish (*Chromis viridis*) was the most imported species, comprising 12.4% of the total number of imported marine ornamental fish, followed by the clown anemonefish (*Amphiprion ocellaris*) with 10.0% and the bicolor angelfish (*Centropyge bicolor*) with 9.4% and of the total number of specimens (Table 4). The 20 most traded species accounted for 63.7% of the overall number of specimens imported into the EU between 2014 and 2021 when the species was known (Table 4, Appendix A). The H’ values concerning the diversity of marine ornamental fish species imported into the EU countries dropped from 4.30 in 2014 to 3.46 in 2021, with E’ values also dropping from 0.59 in 2014 to 0.48 in 2021 (Appendix A).

Of the 20 most traded species, a total of 19 were listed as being of “least concern” by the IUCN Red List, with only the Banggai cardinalfish (*Pterapogon kauderni*) ranked as the ninth most imported species being considered “endangered” (Table 4). It is also worth highlighting that 14 of the 20 most traded species listed were last evaluated ≥10 years ago (Table 4), and three had an IUCN Red List population trend listed as “decreasing” and seven as “unknown” (Table 4).

The IUCN Red List conservation status of all 1,452 species traded between 2014 and 2021 showed 1.3% to be “data deficient” or “not evaluated”, 95.5% as being “least concern”, 2% of species as “endangered”, and only three species with 102 specimens being “critically endangered” (Table 5). The three “critically endangered” species imported were the scalloped hammerhead (*Sphyrna lewini*), which is listed on CITES Appendix II, with 23 specimens being imported in 2015 from the Philippines and destined for France and 75 specimens in 2018, with two from Kenya destined to France, 62 from Australia destined to the Netherlands and 11 from Singapore destined to the Netherlands). Two Nassau groupers (*Epinephelus striatus*) were imported in 2016 from the Philippines and destined for Italy. Sand tiger sharks (*Carcharias taurus*) were imported twice in 2020, both from the USA and destined for the Netherlands (Table 5). No species traded were listed as “extinct in the wild” or “extinct”.

### 3.4. Watchlists

The Watchlist alert system provides a ranking of traded species (where the species is known) from 2014 to 2021 based on a number of trade specimens, the IUCN Red List conservation status, and vulnerability according to FishBase. The first 10 species are either CR or EN, with all species in trade already listed in CITES Appendix II being present in the first 40 species of the Watchlist. Thirty-one species of pipefish and seahorses (Syngnathidae) are on the Watchlist, which is also listed in CITES Appendix II (Table 6; Appendix A).

The most traded species, the blue-green damselfish (*Chromis viridis*), leads the ranking of the WatchlistPLUS, which includes linear regression (slope of number of traded specimens over 8 years). It is followed by the Clown anemonefish (*Amphiprion ocellaris*) and the Copperband butterflyfish (*Chelmon rostratus*). Unlike in the Watchlist, no cartilaginous fish appear on the WatchlistPLUS, as they have either been traded in very low numbers or the number of species traded largely fluctuated over consecutive years, thus making the linear regression not meaningful (Table 7; Appendix A). 

## 4. Discussion

The EU is a major global economic market and a leading importer of marine ornamental fish. In this way, it is legitimate to say that the EU plays a crucial role in ensuring the sustainability of this trade. Indeed, the EU has taken significant steps in this regard, including the “Revised EU Action Plan to End Wildlife Trafficking” [54] and the “European Animals Health Law”, which addresses disease transmission risks in wild animals [55]. Also, the EU, together with the US and Switzerland, asked for this trade to be scrutinized at the 18th conference of the Parties to CITES in August 2019 [56]. In this sense, it is a logical continuation of these actions to adapt TRACES to properly monitor the trade in ornamental marine fish. The EU Parliament’s resolution of 5 October 2022 stressed the importance of addressing the trade of marine ornamental fish. It urged the Commission to modify the European TRACES database, ensuring accurate and publicly accessible information on species, specimen numbers, and trade origins [54]. 

### 4.1. The European Trade Control and Expert System TRACES

Although TRACES is not a monitoring tool for specifically targeting wildlife, it provides valuable data for estimating the number of specimens and species diversity in the trade of marine ornamental fish imported to Europe [7]. TRACES, implemented in the EU in 2004, became applicable for monitoring the marine ornamental fish trade in 2014 [7]. All marine ornamental fish imported to the EU arrive via air freight and undergo customs clearance and veterinary inspection upon arrival. Documents accompanying the shipments provide species-level information, which may be more specific in taxonomic detail from the electronically filled-out TRACES information. This more detailed information can be easily input into TRACES without imposing excessive workload on users since it has been available electronically since 2019 and can be imported directly into TRACES [57]. Moreover, TRACES allows for electronic data import, alignment with FAIR data principles (i.e., data that are findable, accessible, interoperable, and reusable), and accurate scientific identification of fish species. In a survey conducted in 2008, industry representatives expressed support for trade monitoring through veterinary controls, as the forms used already request species-level information and are routinely completed [16]. Unfortunately, the survey yielded no results.

Also, it could potentially be used for monitoring other vertebrate taxa as well, and its fine-tuning for this purpose could easily be tested by using marine ornamental fish as a case study. However, an update in 2019 yielded less accurate data collection for marine ornamental fish, also decreasing data granularity, as it made it possible to input higher taxa (e.g., genus, family or even order). Nonetheless, this updated version is also easier to handle, as data files can be easily imported in the forms of xls and csv database files rather than having to retrieve information by hand over several thousands of physical documents.

It is also worth noting that monitoring systems outside of the EU are often much more inaccurate than TRACES [11,58]. For instance, the USA relies on the Law Enforcement Management Information System (LEMIS), which lacks taxonomic detail and reports most of the trade as generic categories (less than 0.2% or fewer than 22,000 individuals/year at species or genus level) [59]. Australia’s monitoring system also has a very limited taxonomic resolution, leaving uncertainties about the true nature of their imports [58].

Extracting information based on weight (kg/year) from databases such as UN Comtrade [40] has been suggested as a proxy for monitoring this trade, but this approach includes using the weight of water in which fish are shipped, thus making this figure highly unreliable for determining the number of marine ornamental fish being imported, exported, and re-export [59]. The lack of species-specific information not only hampers scientific analysis but also poses a biosecurity risk, which the EU aims to address through its Animal Health Law [55]. Australia recognizes the risks associated with the trade of ornamental fish and has implemented strict import biosecurity measures to control diseases [60,61].

### 4.2. Import Values

According to European imports and total demand, marine ornamental fish are becoming more expensive. The global trade of marine ornamental fish has always been much more valuable than that of food fish [49]. In the 1980s, marine aquarium fish were priced at USD 750/kg, while marine food fish were priced at USD 9/kg [62]. Presently, marine ornamental fish are valued at a minimum of USD 1000/kg compared with USD 13/kg for marine food fish [63]. It had previously been estimated that globally, coral reefs contribute USD 2.7 trillion annually in goods and services, including USD 36 billion in coral reef tourism [32,64]. The industry of aquatic organisms for home and public aquariums, along with the equipment required to display these organisms, is estimated to be a multi-billion-dollar industry [65]. In the 1980s, the global ornamental fish industry with associated equipment and accessories was valued at USD 7.2 billion [66], which increased to USD 20–30 billion by 1997 [5,64,67,68,69,70]. By 2004, the estimated value ranged from USD 800 million to USD 30 billion annually [5,65,71,72,73]. Currently, the import value of marine ornamental fish solely shipped to and within Europe is EUR 24.7 million, while total demand is about EUR 37.6 million. On the other hand, extra-EU imports diminished from EUR 13 million to EUR 9.5 million between 2014 and 2021. The latter figure is similar to extra-EU import estimates previously reported for the period 2000–2011 [4]. This figure is likely to be more comparable to the import levels of regions where intra-regional trade is not accounted for in the data (especially in the case of China in this analysis). Although the number of specimens recorded by TRACES has decreased, the value of intra-EU imports into the EU, as well as the value of total demand, has remained steady. The long-term trend of the EU final demand with regards to value does not appear to be diminishing, and therefore, the reduction in the number of specimens may be a result of the market moving into “higher-value market niches” or the result of a rigid supply being unable to meet demand (as the increase in average prices suggests). The sharp increase in European average prices in 2021 accelerated the increasing trend in prices, most likely as a result of the COVID-19 pandemic. This emphasizes the need for improved trade reporting and monitoring systems, as urged by other studies [74]. Overall, fewer specimens per year were imported to Europe from 2014 to 2021 at an almost constant decreasing rate. As the import value stayed the same over the eight years, each fish specimen became more expensive. A reason for a product to become more expensive is if the supply becomes scarcer, often due to a population decrease when it comes to live specimens. Another explanation for an increase in price would be a higher demand driven by other countries, such as China, and supply being unable to keep up.

Increasing import values in non-EU countries, mainly China, the USA, and the Middle East, suggest that the international demand for marine ornamental fish will continue to expand, pushing prices up and increasing the pressure on local fish populations. It is, therefore, crucial to recognize that live marine ornamental fish are just one part of a larger commercial system that includes food, supplements, and a wide range of accessories and equipment. Unequivocally, the value of this system is much greater than that of live marine ornamental fish alone [9] and should be addressed in future studies. The extensive economic impact of this integrated trade, supported by a complex network of suppliers, traders, retailers, and consumers, puts significant pressure on marine ornamental fish used as ornamentals. 

### 4.3. Assessment of Environmental Consequences 

Habitat loss is the greatest threat to biodiversity [75], and coral reefs are among the most threatened marine ecosystems due to anthropogenic interferences [31,32,76]. The Intergovernmental Science-Policy Platform on Biodiversity and Ecosystem Services (IPBES) has identified overexploitation, including trade, as the second leading cause of extinction for nearly one million species [77]. Estimates suggest that global trade involves 15–30 million coral reef fish annually [9], with potential figures reaching as high as 150 million [65]. Marine ornamental fish have yet to be included in CITES Appendix I (trade ban), with only seahorses, humphead wrasse (*Cheilinus undulatus*), and clarion angelfish (*Holocanthus clarionensis*), as well as a few sharks and rays, monitored under CITES Appendix II (monitored trade). 

Almost all marine ornamental fish traded worldwide are still sourced from the wild, mostly from coral reefs, as they have complex life cycles that are difficult to replicate in breeding facilities or aquariums [5,9,11,12,78,79]. As of 2018, only 24 species were bred in commercial numbers in captivity, and 37 were moderately available, while another 277 had already been bred at what can be considered a more research level; indeed, the number of species bred in captivity has remained fairly constant since 2012 [79]. Keeping marine ornamental fish is expensive and demanding [80]. Furthermore, a survey of over 3000 marine ornamental fish keepers showed that more than 70% did not intend to try to breed marine ornamental fish [81].

### 4.4. Origin, Destination, Diversity, and Conservation

Overall, from 2014 to 2021, 61 countries exported marine ornamental fish to the EU, which represents an increase of 18% from 2014 to 2017 [5]. Some small island countries, such as Palau, São Tomé and Príncipe, and Tonga, did not appear to export to the EU in the last few years of this analysis despite having rare and highly sought-after species for the marine aquarium trade [57]. Singapore still plays a significant role as a transit hub for this trade, and it remains unclear where species labeled as originating from Singapore were indeed collected in that country. It is important to note that China’s domestic wildlife trade is significant, emphasizing the need for global monitoring. The Netherlands surpassed the UK as the leading importer in 2021 after the latter exited the EU (an event popularly known as Brexit). The decreasing trend recorded for H’ and E’ values suggests that fewer countries are exporting to EU countries and that some countries may have enhanced their prevalence as suppliers of marine ornamental fish into this market, respectively.

Species diversity (number) of imported marine ornamental species reach a total of 1452 species over the whole period. Unfortunately, for one-third of all specimens traded, it was not possible to identify them at the species level. Strangely, a CITES document analyzing TRACES data from 2018 to 2021 found only 33 species in TRACES [59], with some species using outdated names no longer accepted by WoRMS (e.g., starry triggerfish, *Abalistes stellaris* instead of *A. stellatus*) [82].

A total of 120 families were imported over the period surveyed, increasing steadily from the lowest value record in 2017 at 64 families to the highest one in 2021 at 90 families. This rise may suggest that the most commonly available species in the trade may no longer be so abundant in the wild due to the dire state of the marine ornamental fish’ habitats—tropical coral reefs [29,31,32]. Nonetheless, H’ and E’ values have remained constant over the years. Hence, additional data must be collected and consolidated for subsequent years to detect if there is indeed a pronounced increase in the number of marine ornamental fish families being targeted by the global marine aquarium trade.

The number of the most traded species, the blue-green damselfish *Chromis viridis*, has diminished significantly by 70%. The species was last evaluated in 2021 and is listed as “least concern”, but the wild population is decreasing [83]. Also, the second most traded species, the clown anemonefish *Amphirion ocellaris*, showed a decrease of almost 70% in eight years. This species was also last evaluated in 2021 and is of “least concern” according to the IUCN Red List, although its population status is unknown [84]. These two species account for almost a quarter of the whole trade into the EU. While *A. ocellaris* is increasingly bred in captivity in importing countries, and this may somewhat explain why fewer specimens are being imported into the EU, this is certainly not the case for *C. viridis* [79]. The third most traded species is the bicolor angelfish (*Centropyge bicolor*), which, according to the IUCN Red List, has a stable population but was last evaluated in 2009 [85]. The marine aquarium trade regularly introduces new fish species to meet the demand for unique and novel organisms sought by hobbyists, an approach that aims to keep the price of “rare” species in the high-end range [86]. This practice suggests that an increase in species diversity presented in the trade could be anticipated unless one species previously offered in the trade is replaced by a new one. Additionally, one cannot discard aquarium hobbyists’ “fashion and trends”, as these will also play an important role in shaping the trade of marine ornamental fish. The values recorded for H’ in 2020 and 2021 showed a pronounced decrease, thus indicating that fewer species of marine ornamental fish were being imported into the EU. The decrease recorded in E’ values also suggests that the volume of imported specimens is being concentrated in fewer species, hence decreasing the evenness of contribution of all species traded to the overall number of specimens imported into the EU.

Our study allowed us to confirm that some “critically endangered” fish species were imported (see above). For instance, in 2015, France imported 23 scalloped hammerheads *Sphyrna lewini* (listed on CITES Appendix II) from the Philippines; in 2016, Italy imported two “critically endangered” Nassau groupers *Epinephelus striatus* from the Philippines, and in 2018, 75 specimens of scalloped hammerheads *Sphyrna lewini* were also imported: two from Kenya destined to France, 62 from Australia destined to the Netherlands, and 11 more from Singapore also destined to the Netherlands. Three hammerhead sharks, recorded only at the family level, were imported to Italy from Sri Lanka. Sand tiger sharks *Carcharias taurus* were imported twice in 2020, both from the United States to the Netherlands, reflecting the presence of a major shark wholesaler in the country equipped with facilities to hold such large and active animals. Notably, eleven scalloped hammerheads had Singapore as their country of origin, highlighting Singapore’s role as a transit hub in the marine aquarium trade. This increase in live-shark trade is concerning, as nearly two-thirds of shark and ray species associated with coral reefs are at risk of extinction [87] due to the ongoing decline of shark species caused by commercial fishing and by-catch, despite all management efforts to revert this trend [88].

Of the 1452 marine ornamental fish species traded between 2014 and 2021, 1.3% had a conservation status of “data deficient” or “not evaluated”. This is a stark decrease compared with 33.63% of the 1,334 species traded (with known species level) between 2014 and 2017 [5]. While it is positive that fewer species require evaluation, the IUCN Red List advises not to assume that these categories indicate non-threatened status. Until assessed, it is recommended to give the same attention to data-deficient species as to already recognized threatened species [89]. Furthermore, for many species of marine ornamental fish, the IUCN Red List evaluation is outdated by over ten years, leaving their current conservation status unclear. It is possible that “data deficient” species may be more threatened than initially perceived [90]. Research indicates that a third of species listed as “not threatened” are experiencing a decline [88]. Moreover, for nearly 75% of the 25,000 analyzed fish species, the population trend is unknown due to data limitations, which poses a significant challenge to understanding their status [91]. 

### 4.5. Watchlist and WatchlistPLUS Alert Systems

The Watchlist considers the number of traded specimens, species’ vulnerability according to Fishbase, and the IUCN Red List conservation status and gives a ranking using the overall scores, while the WatchlistPLUS further considers a linear regression of the number of specimens being annually traded. Both can be used as an indicator of potential negative impacts promoted by the international trade of marine ornamental species, with the WatchlistPLUS having more robust statistics regarding the trend of specimens traded. Both watchlists can serve as an alert system, a starting point to identify species that require closer analysis or observation, whether due to the high number of specimens being traded or their ecological vulnerability that may lead to a possible population decline. Moreover, it also provides insights into which species could receive better precautionary monitoring through CITES. Unfortunately, about a third of all imported marine ornamental fish specimens lack species identification and, therefore, could not be taken into consideration to enhance the magnitude of this watchlist approach. If up-to-date information is made available on the conservation status of the most traded marine ornamental fish species, watchlists may allow policymakers to take action before reaching tipping points, and fisheries need to be closed.

In the Watchlist, the blue-green damselfish (*Chromis viridis*) ranked 22nd, representing the first Osteichthyes in the Watchlist but first in the rankings in the WatchlistPLUS due to its large numbers in trade and the strongest decline recorded in the number of traded specimens. The blue-green damselfish, initially considered a complex of two species (*C. viridis* and *Chromis atripectoralis*), were differentiated primarily based on the coloration of the pectoral fin base [92]. Interestingly, *C. atripectoralis* was mainly exported from the Philippines, while imports of *C. viridis* listed the Caribbean region (e.g., Cuba) as the country of origin. However, the Caribbean region is not a natural habitat for this Indo-Pacific species. Both species leading the WatchlistPLUS show a pronounced decline in the number of traded specimens [89] and qualify for an in-depth analysis of the reasons why such a trend was displayed. 

As the linear regression was not meaningful, the Banggai cardinalfish (*Pterapogon kauderni*) ranked 36th on the Watchlist but did not make it on the WatchlistPLUS. In contrast to the Watchlist, no cartilaginous fish, even if already listed on CITES Appendix II, made it to the WatchlistPLUS, as they have been either traded in very low numbers or the number of species traded largely fluctuated over consecutive years. Similarly, the humphead wrasse (*Cheilinus undularus*), which is also listed in CITES Appendix II, also did not make it to the WatchlistPLUS due to the reduced numbers in trade, most likely due to its large size at adulthood, thus only being viable for display on public aquariums. Nonetheless, as over half a million wrasses (Labridae) were not registered at the species level, the accuracy of this finding must be put into perspective, reinforcing the need for a more accurate monitoring system. 

### 4.6. Advantages of TRACES and Its Adaptation to Monitor Wildlife

The trade of marine ornamental fish can pose biosecurity risks, potentially leading to the unintentional spread of pathogens such as viruses [60,61]. Additionally, there is a hazard of introducing exotic species that may become invasive, exemplified by the red lionfish (*Pterois volitans*). Despite their negative impacts, such as predation on smaller fish and the establishment of large populations, *P. volitans* specimens are still imported into the USA [93]. The EU should take heed of this lesson and prohibit *P. volitans* imports, as *Pterois* species are already present in the Mediterranean Sea [94]. While the import of the devil firefish (*Pterois miles*) was banned in 2015, over 300 specimens have been imported into the EU since 2016 [95].

The absence of species-level information in the long-standing marine ornamental fish trade has significant conservation implications, as highlighted in previous studies [5,8,9,13,59,78].

## 5. Conclusions

Globally, the EU is the largest import market for marine ornamental fish by value and hence carries an added responsibility to promote sustainable management of this commercial activity. While TRACES is not specifically designed for monitoring trade in wildlife, including marine ornamental fish, it already possesses the necessary features for such purposes, and the pet trade industry has already endorsed its use [16]. TRACES can be easily adapted to collect accurate data independently of CITES decisions, as traders are already required to submit information electronically, and as an EU stakeholder survey suggests, they would be willing to do so [96].

TRACES should be fine-tuned so it would (a) only allow marine ornamental fish under code “HS 03011900”, i.e., coral reef fish according to the World Register of Marine Species (WoRMS; www.marinespecies.org (accessed on 23 September 2023)); (b) only accept scientific species names; (c) make it mandatory to specify if specimens being traded were sourced from the wild by specifying geographic origin (country of capture and region within the country of capture) or if they were captive bred by providing the address of the breeding facility.

With the WatchlistPLUS incorporating a linear regression model to assess the temporal trend in the volume of specimens traded, the EU can prioritize the survey of species potentially vulnerable to overexploitation as a result of the global marine aquarium trade. Moreover, a fine-tuned TRACES may also contribute to clarifying the potential confounding effects on the imports of marine ornamental fish into the EU, which will be promoted by both “Brexit” and the COVID-19 pandemics in the future. Overall, a refined TRACES will be a valuable tool for accurately reporting reliable data on the trade of wildlife in general and live marine ornamental fish in particular in the EU. 

## Figures and Tables

**Figure 1 animals-14-01761-f001:**
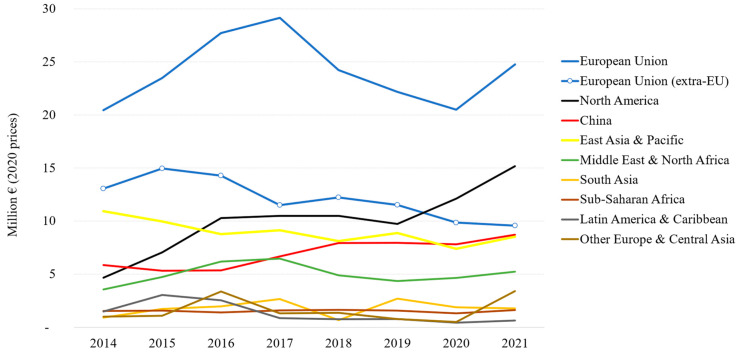
Lobal imports by region 2014–2021 in million EUR (2020 prices). Notes: (1) The European Union (EU) includes 27 EU countries plus the UK (until 2020), Switzerland, Norway, Iceland, and San Marino (The United Nations Comtrade (UN Comtrade) database, code “HS 03011900 Live ornamental fish (excluding freshwater)”. (2) Import values are deflated by the Harmonized Index of Consumer Prices (EuroStat 2023) [44].

**Figure 2 animals-14-01761-f002:**
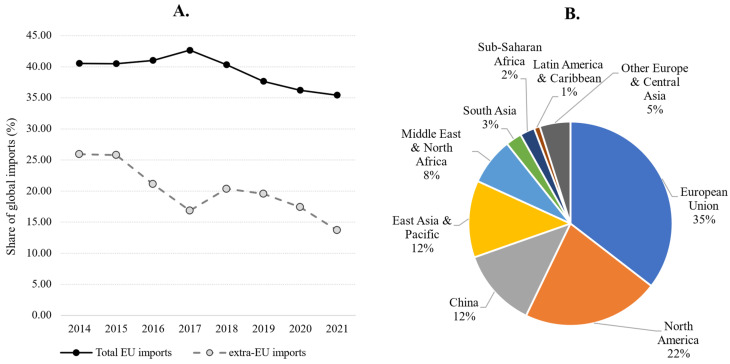
(**A**): Market share of European imports in the global imports value of marine ornamental fish. (**B**): Share (%) of import value by region in 2021. Source: Own calculations based on UN Comtrade database, “HS classification 03011900 Live ornamental fish (excluding freshwater)”.

**Figure 3 animals-14-01761-f003:**
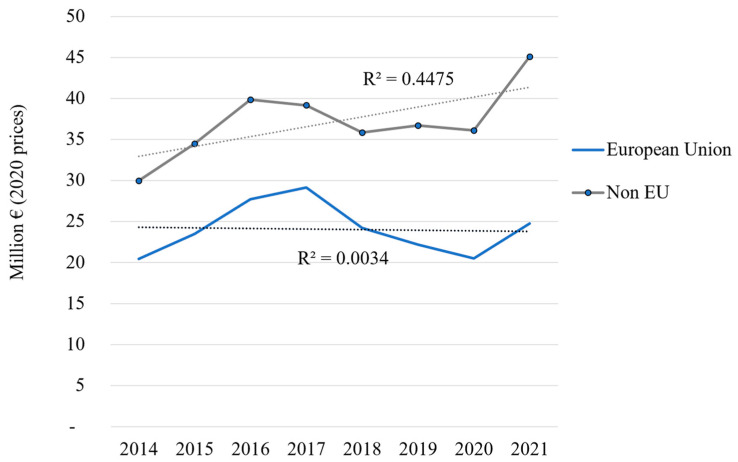
Global trade and linear regression (dotted line) in marine ornamental fish: annual imports by region 2014–2021 (million EUR at 2020 prices); own calculations based on UN Comtrade database, classification “HS 03011900 Live ornamental fish (excluding freshwater)”.

**Figure 4 animals-14-01761-f004:**
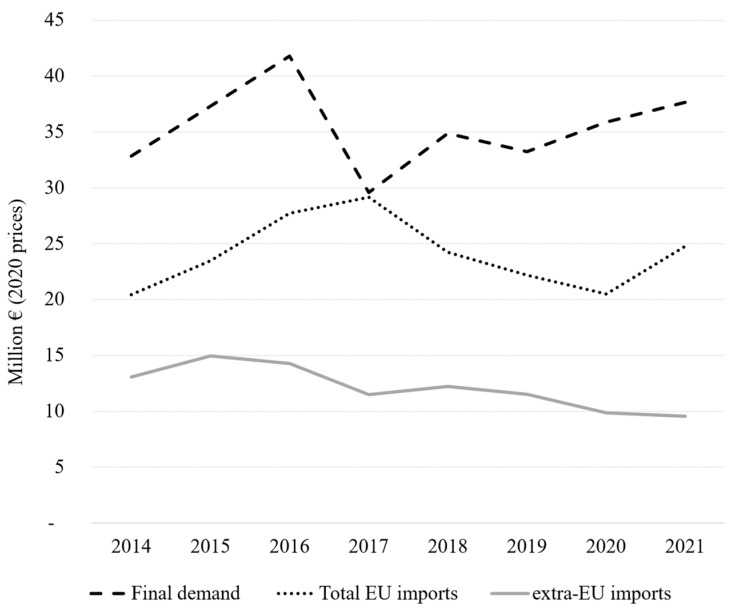
European value trade (imports) and final demand in marine ornamental fish (million EUR at 2020 prices) from 2014 to 2021; authors’ calculations based on UN Comtrade database, classification “HS 03011900 Live ornamental fish (excluding freshwater)”.

**Figure 5 animals-14-01761-f005:**
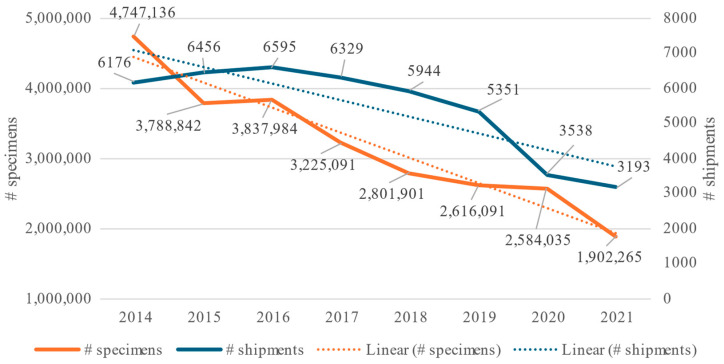
Overall shipments and number of traded specimens per year of marine ornamental fish and linear regression from 2014 to 2021 entering Europe according to data from the TRAde Control and Expert System (TRACES).

**Figure 6 animals-14-01761-f006:**
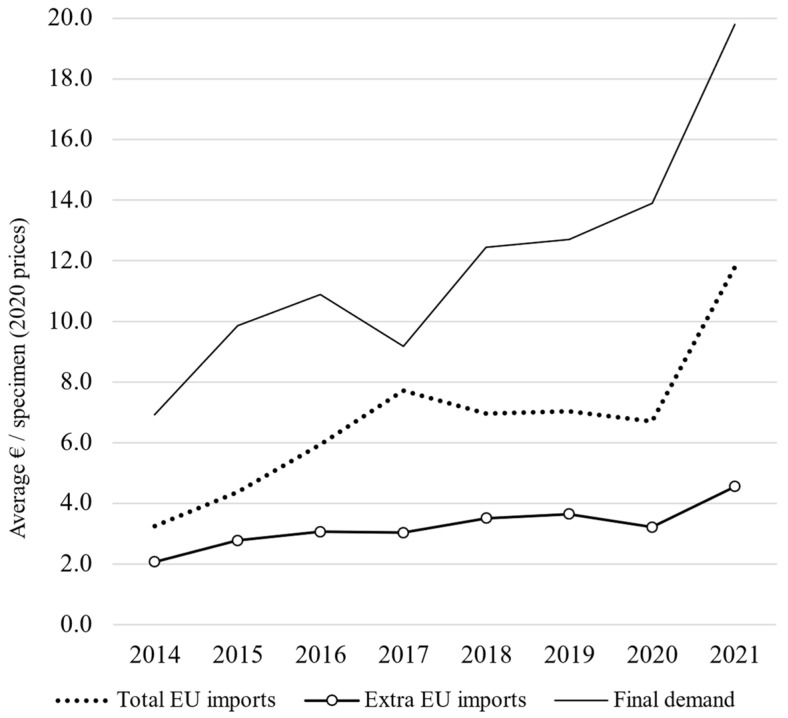
European marine ornamental imports: average prices and final demand in marine ornamental fish (value in EUR, 2020 prices) from 2014 to 2021; authors’ calculations based on the EuroStat database and TRACES data.

**Figure 7 animals-14-01761-f007:**
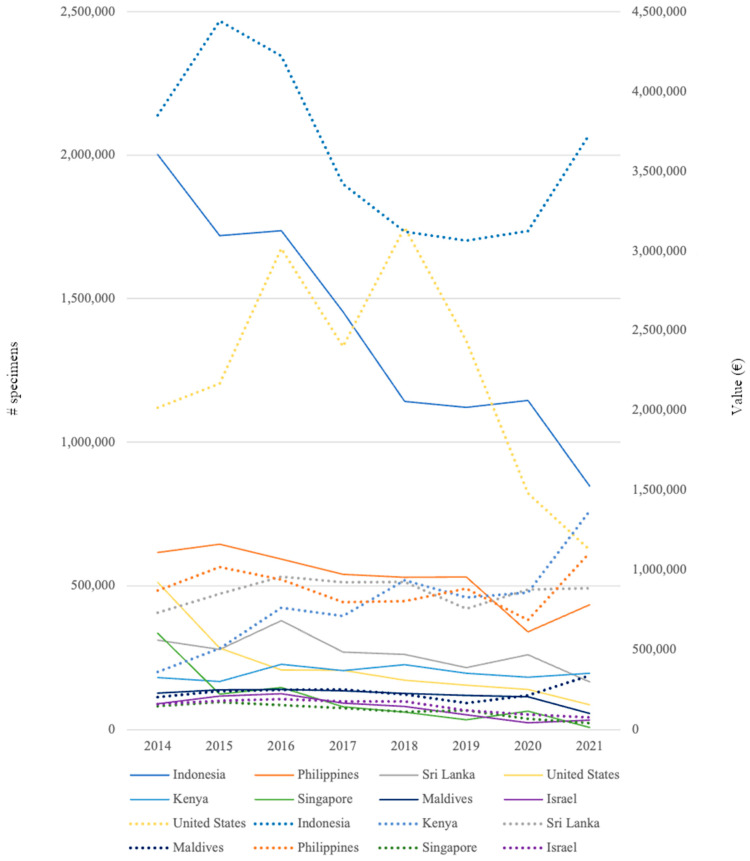
Number of specimens of marine ornamental fish traded and value per year of the top eight exporting countries from 2014 to 2021 entering Europe according to TRACES data. Straight line = specimens; dotted line = value (EUR).

**Figure 8 animals-14-01761-f008:**
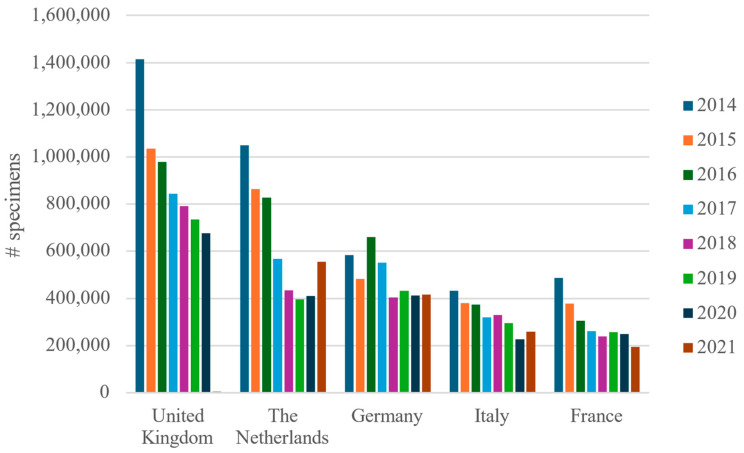
Number of specimens of marine ornamental fish imported to the five top importing European countries by year from 2014 to 2021, according to TRACES data. The United Kingdom left the EU at the end of 2020.

**Figure 9 animals-14-01761-f009:**
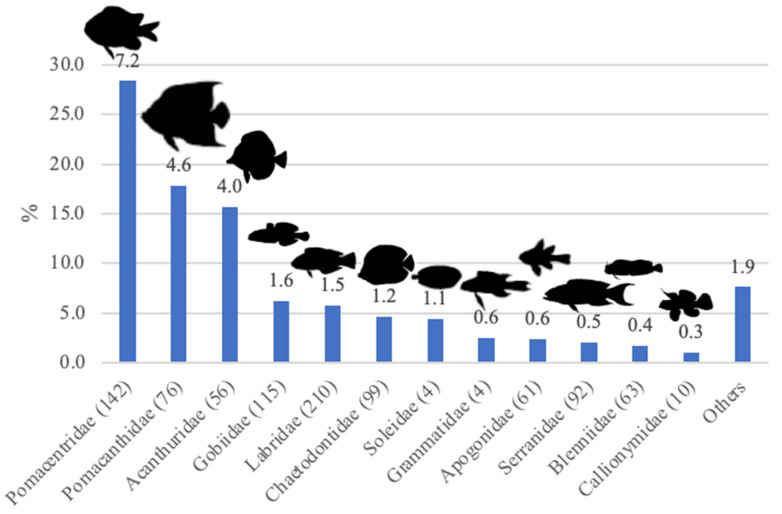
Number of specimens of marine ornamental fish (% of total imported specimens) of the top twelve families traded into Europe between 2014 and 2021, according to TRACES data. The number of imported species per family is presented in parentheses, with the number on top of the bar representing millions of specimens.

**Table 1 animals-14-01761-t001:** Top eight exporting countries of marine ornamental fish to Europe between 2014 and 2021. Average, standard deviation, and % of shipments and specimens and value of imports per year. Total imports refer to imports from outside the EU 27. AVG = average, SD = standard deviation. Value of imports, EUR (2020 prices). Total = extra-EU 27 imports, excluding intra-EU trade; EuroStat [44].

	Shipments			Specimens			Value of Imports (EUR)		
Country of Origin	AVG	SD	%	AVG	SD	%	AVG	SD	%
Indonesia	1823.6	436.8	33.5	1,394,208	393,487	43.7	3,621,793	527,884	38.1
Philippines	663.5	183.6	12.2	529,076	100,105	16.6	888,489	133,226	9.3
Sri Lanka	883.3	175.6	16.2	266,945	62,842	8.4	863,273	80,583	9.1
United States	441.8	162.6	8.1	221,012	131,618	6.9	1,235,212	384,862	13.0
Kenya	391.0	74.7	7.2	197,073	21,001	6.2	791,417	300,613	8.3
Maldives	198.8	66.9	3.6	118,709	26,912	3.7	234,698	50,275	2.5
Singapore	356.8	123.4	6.5	107,222	103,373	3.4	118,618	44,016	1.3
Israel	236.6	91.8	4.3	86,151	31,104	2.7	146,374	43,452	1.5
Fiji	58.3	23.9	1.1	46,633	21,231	1.5	51,124	15,433	0.5
Dominican Republic	57.5	28.1	1.1	37,638	31,289	1.2	26,789	18,208	0.3
Other	12.4		6.2	6756		5.7	1,531,137		16.1
Total	5123.5		100.0	3,011,423		100.0	9,508,924		100.0

**Table 2 animals-14-01761-t002:** Average number of specimens per shipment and average price per specimen in EUR (2020 prices) of the top eight exporting countries between 2014 and 2021; EuroStat [44].

Country of Origin	Specimens/Shipment	EUR/Specimen
Indonesia	765	2.6
Philippines	797	1.7
Sri Lanka	302	3.2
United States	333	5.6
Kenya	504	4.0
Maldives	597	2.0
Singapore	301	1.1
Israel	364	1.7
Fiji	801	1.1
Dominican Republic	655	0.7

**Table 3 animals-14-01761-t003:** Number of imported marine ornamental fish per European country between 2014 and 2021, with a total number of specimens over eight years. NA = not applicable.

Country of Origin	2014	2015	2016	2017	2018	2019	2020	2021	Total
United Kingdom	1,414,494	1,034,680	978,506	844,667	791,320	734,017	677,486	5589	6,480,759
Netherlands	1,049,798	864,511	828,314	567,534	434,157	397,051	410,299	555,299	5,106,963
Germany	583,889	482,637	660,114	551,266	404,108	432,017	411,755	416,326	3,942,112
Italy	431,839	379,421	373,391	320,154	328,957	295,946	226,673	258,509	2,614,890
France	487,863	377,900	305,278	261,525	240,181	257,648	249,002	195,783	2,375,180
Spain	237,842	117,045	131,727	81,042	129,689	102,110	49,573	46,768	895,796
Poland	93,191	78,044	87,300	94,473	85,277	96,609	119,399	106,882	761,175
Denmark	68,144	120,674	131,517	157,571	66,279	43,025	150,020	11,731	748,961
Belgium	99,186	62,984	76,810	70,276	59,691	48,769	67,871	39,246	524,833
Sweden	30,922	48,561	42,061	81,367	33,404	24,106	31,255	12,912	304,588
Switzerland	38,407	37,217	42,152	43,138	34,650	29,855	27,732	28,298	281,449
Czech Republic	22,187	27,462	26,633	21,817	41,022	31,291	60,351	34,426	265,189
Portugal	10,067	23,633	23,900	17,178	30,136	11,809	28,998	118,660	264,381
Austria	33,803	34,032	35,225	34,999	36,200	23,270	19,752	24,490	241,771
Greece	45,573	31,876	26,898	21,880	22,383	16,707	13,275	16,538	195,130
Norway	48,171	26,843	33,440	20,321	17,116	12,766	3507	7244	169,408
Hungary	20,442	7065	6834	9797	9670	9795	11,049	7044	81,696
Luxembourg	13,311	12,248	10,319	6683	6536	7339	1958	NA	58,394
Romania	1650	1344	1613	6124	19,121	13,882	6774	523	51,031
Cyprus	6576	6068	4735	5566	4102	13,926	4138	4339	49,450
Malta	4992	10,130	5530	5476	3835	9943	1786	1839	43,531
Bulgaria	2517	766	909	1681	2333	1637	419	887	11,149
Ireland	838	190	826	175	397	281	2313	6005	11,025
Other	NA	NA	NA	NA	NA	Na	6983	NA	6983
Croatia	481	879	519	199	958	1584	1072	1270	6962
San Marino	NA	1801	2392	NA	NA	NA	NA	NA	4193
Slovenia	953	770	1041	182	144	NA	NA	NA	3090
Iceland	NA	NA	NA	NA	235	213	595	1657	2700
Estonia	NA	NA	NA	NA	NA	399	NA	NA	399
Finland	NA	NA	NA	NA	NA	96	NA	NA	96
Slovakia	NA	61	NA	NA	NA	NA	NA	NA	61

**Table 4 animals-14-01761-t004:** Top 20 species of marine ornamental fish imported to Europe between 2014 and 2021 and their IUCN Red List conservation status (LC = “least concern”, EN = “endangered”), population trend, and year assessed. AVG = average, SD = standard deviation, % = percentage of traded specimens.

Species	Family	Common Name	IUCN Red List Status	IUCN Population Trend	Year Assessed	2014	2015	2016	2017	2018	2019	2020	2021	Total	AVG	SD	%
*Chromis viridis*	Pomacentridae	Blue-green damselfish	LC	decreasing	2021	334,458	377,022	385,430	348,661	308,112	197,159	146,548	114,370	2,211,760	276,470	107,573.7	12.4
*Amphiprion ocellaris*	Pomacentridae	Clown anemonefish	LC	unknown	2021	257,520	370,199	344,363	201,973	198,505	157,504	117,887	120,680	1,768,631	221,079	95,923.0	10.0
*Centropyge bicolor*	Pomacanthidae	Bicolor angelfish	LC	stable	2009	187,298	258,010	265,842	251,070	190,944	200,712	150,247	173,400	1,677,523	209,690	43,068.1	9.4
*Acanthurus leucosternon*	Acanthuridae	Powderblue surgeonfish	LC	unknown	2010	105,389	96,610	99,005	86,144	93,701	85,862	95,718	3727	666,156	83,270	32,777.7	3.7
*Chelmon rostratus*	Chaetodontidae	Copperband butterflyfish	LC	stable	2009	124,892	137,841	85,396	48,954	50,026	117,381	62,497	6079	633,066	79,133	45,378.2	3.6
*Gramma loreto*	Grammatidae	Royal gramma	LC	unknown	2011	130,850	75,770	55,206	80,675	101,047	87,203	59,132	21,920	611,803	76,475	32,564.3	3.4
*Pomacanthus imperator*	Pomacanthidae	Emperor angelfish	LC	stable	2009	64,740	53,493	27,088	26,183	95,945	142,353	93,730	105,073	608,605	76,076	40,474.9	3.4
*Acanthurus achilles*	Acanthuridae	Achilles tang	LC	stable	2010	84,317	87,950	83,788	105,511	596	3768	4166	1500	371,596	46,450	47,466.2	2.1
*Pterapogon kauderni*	Apogonidae	Banggai cardinalfish	EN	decreasing	2007	43,982	38,169	56,494	56,649	42,557	38,374	40,604	33,058	349,887	43,736	8573.5	2.0
*Zebrasoma flavescens*	Acanthuridae	Yellow tang	LC	stable	2010	40,959	54,527	54,535	45,846	42,191	27,730	10,258	1212	277,258	34,657	19,905.1	1.6
*Centropyge bispinosa*	Pomacanthidae	Twospined angelfish	LC	stable	2009	37,520	32,310	23,273	65,186	44,313	49,152	23,661	1664	277,079	34,635	19,228.4	1.6
*Labroides dimidiatus*	Labridae	Bluestreak cleaner wrasse	LC	unknown	2008	28,988	46,030	22,903	60,396	31,926	35,948	15,339	15,800	257,330	32,166	15,354.9	1.4
*Valenciennea sexguttata*	Gobiidae	Sixspot goby	LC	unknown	2017	8015	34,192	81,751	85,092	28,305	9653	4007	5510	256,525	32,066	33,535.4	1.4
*Chrysiptera parasema*	Pomacentridae	Goldtail demoiselle	LC	decreasing	2021	68,649	60,702	33,348	32,110	20,177	26,339	10,289	2374	253,988	31,749	22,957.0	1.4
*Valenciennea puellaris*	Gobiidae	Maiden goby	LC	unknown	2015	22,715	33,845	23,828	39,997	41,066	31,831	42,649	11,939	247,870	30,984	10,758.1	1.4
*Centropyge eibli*	Pomacanthidae	Blacktail angelfish	LC	stable	2009	35,204	12,473	7384	34,345	13,976	11,704	72,292	1508	188,886	23,611	23,083.6	1.1
*Paracanthurus hepatus*	Acanthuridae	Palette surgeonfish	LC	unknown	2010	19,735	17,409	48,129	26,761	14,553	19,294	23,123	2617	171,621	21,453	12,922.5	1.0
*Salarias fasciatus*	Blenniidae	Jewelled blenny	LC	stable	2009	19,465	14,728	22,932	38,089	25,479	18,158	15,008	11,643	165,502	20,688	8351.9	0.9
*Amphiprion percula*	Pomacentridae	Orange clownfish	LC	stable	2010	11,290	16,685	22,113	36,323	27,624	14,798	15,232	20,142	164,207	20,526	8137.1	0.9
*Pseudanthias squamipinnis*	Serranidae	Sea goldie	LC	stable	2015	20,229	19,840	28,918	28,896	23,646	22,616	9416	10,605	164,166	20,521	7338.8	0.9

**Table 5 animals-14-01761-t005:** IUCN Red List conservation status of all species of marine ornamental fish and number of specimens per species imported into Europe between 2018 and 2021, as well as their % of the number of known marine ornamental fish species in Europe and worldwide, according to FishBase. NE = “not evaluated”, DD = “data deficient”, LC = “least concern”, NT = “near threatened”, VU = “vulnerable”, EN = “endangered”, CR = “critically endangered”.

IUCN Red List Status	2014	2015	2016	2017	2018	2019	2020	2021	Total	%	# Species
NE	17,671	26,354	16,538	12,713	11,412	15,027	8092	12,341	120,148	0.7	63
DD	19,623	33,136	9779	10,315	14,308	10,909	7436	5027	110,533	0.6	53
LC	2,754,733	2,697,184	2,643,394	2,399,028	2,121,468	1,936,373	1,578,146	911,066	17,041,392	95.5	1274
NT	5704	4181	3114	2465	595	1537	683	300	18,579	0.1	18
VU	53,851	34,234	18,156	30,120	17,284	11,946	19,723	15,029	200,343	1.1	31
EN	44,000	38,195	56,526	56,686	42,570	38,421	49,807	33,088	359,293	2.0	10
CR	NA	23	2	NA	75	NA	2	NA	102	0.0	3

**Table 6 animals-14-01761-t006:** First 40 species with their IUCN Red List conservation status on the Watchlist ranked by the sum of three normalized parameters according to TRACES data from 2014 to 2021: the score of the median number of specimens traded, the score of the IUCN Red List conservation status (www.iucnredlist.org (accessed on 23 September 2023)), and the score in vulnerability according to FishBase (http://www.fishbase.org (accessed on 23 September 2023)), resulting in overall score. * CITES Appendix II-listing. 
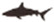

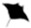
 = Chondrichthyes (sharks and rays), 
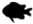
 = Osteichthyes (bony fish).

Rank	Species	Common Name	Family	IUCN Red List Status	Overall Score	Median Volume Score	IUCN Score	Vulnerability Score
1	*Sphyrna lewini ** 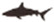	Scalloped hammerhead	Sphyrnidae	CR	158	0	80	78
2	*Himantura uarnak* 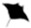	Honeycomb stingray	Dasyatidae	EN	150	0	60	90
3	*Stegostoma tigrinum* 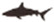	Zebra shark	Stegostomatidae	EN	150	0	60	90
4	*Carcharhinus plumbeus ** 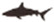	Sandbar shark	Carcharhinidae	EN	148	1	60	87
5	*Rhinoptera javanica* 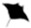	Flapnose rays	Rhinopteridae	EN	146	0	60	86
6	*Carcharhinus amblyrhynchos ** 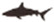	Blacktail reef shark	Carcharhinidae	EN	145	0	60	85
7	*Epinephelus striatus* 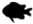	Nassau grouper	Serranidae	CR	143	0	80	63
8	*Carcharias taurus* 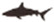	Sand tiger shark	Carchariidae	CR	138	0	80	58
9	*Aetobatus narinari* 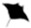	Pacific eagle ray	Aetobatidae	EN	135	0	60	75
10	*Cheilinus undulatus ** 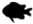	Huamphead wrasse	Labridae	EN	134	0	60	74
11	*Epinephelus lanceolatus* 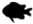	Giant grouper	Serranidae	DD	130	0	40	90
12	*Nebrius ferrugineus* 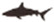	Tawny nurse shark	Ginglymostomatidae	VU	130	0	40	90
13	*Pateobatis jenkinsii* 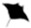	Jenkins whipray	Dasyatidae	VU	130	0	40	90
14	*Taeniurops meyeni* 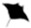	Round ribbontail ray	Dasyatidae	VU	130	0	40	90
15	*Ginglymostoma cirratum* 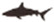	Nurse shark	Ginglymostomatidae	VU	130	0	40	90
16	*Urogymnus asperrimus* 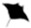	Porcupine whipray	Dasyatidae	VU	130	0	40	90
17	*Aetobatus ocellatus* 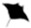	Ocellated eagle ray	Aetobatidae	VU	126	0	40	86
18	*Triaenodon obesus* 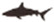	Whitetip reef shark	Carcharhinidae	VU	123	0	40	83
19	*Sphyrna tiburo* 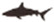	Bonnethead	Sphyrnidae	EN	117	0	60	57
20	*Plectorhinchus albovittatus* 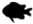	Two-striped sweetlips	Haemulidae	NE	113	0	40	73
21	*Heterodontus francisci* 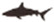	Horn shark	Heterodontidae	DD	113	0	40	73
22	*Chromis viridis* 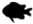	Blue-green damselfish	Pomacentridae	LC	110	100	0	10
23	*Mycteroperca interstitialis* 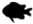	Yellowmouth grouper	Serranidae	VU	107	0	40	67
24	*Sebastes pinniger* 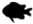	Canary rockfish	Sebastidae	NE	102	0	40	62
25	*Myrichthys maculosus* 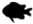	Tiger snake eel	Ophichthidae	NE	100	0	40	60
26	*Chiloscyllium punctatum* 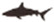	Brownbanded bamboo shark	Hemiscylliidae	NT	99	0	20	79
27	*Myrichthys colubrinus* 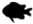	Harlequin snake eel	Ophichthidae	NE	99	0	40	59
28	*Epinephelus fuscoguttatus* 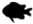	Brown-marbled grouper	Serranidae	VU	97	0	40	57
29	*Carcharhinus melanopterus* 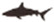	Blacktip reef shark	Carcharhinidae	VU	97	0	40	57
30	*Hypanus americanus* 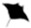	Southern stingray	Dasyatidae	NT	97	0	20	77
31	*Carcharhinus limbatus* 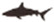	Blacktip shark	Carcharhinidae	VU	95	0	40	55
32	*Callechelys marmorata* 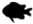	Marbled snake eel	Ophichthidae	NE	95	0	40	55
33	*Cebidichthys violaceus* 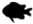	Monkeyfaces prickleback	Cebidichthyidae	NE	94	0	40	54
34	*Neotrygon kuhlii* 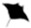	Blue-spotted stingray	Dasyatidae	DD	92	0	40	52
35	*Pomacanthus imperator* 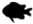	Emperor angelfish	Pomacanthidae	LC	92	25	0	68
36	*Pterapogon kauderni* 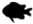	Banggai cardinalfish	Apogonidae	EN	92	13	60	19
37	*Platax batavianus* 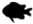	Humpback batfish	Ephippidae	NE	92	0	40	52
38	*Amblyraja radiata* 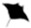	Starry ray	Rajidae	VU	91	0	40	51
39	*Hemiscyllium hallstromi* 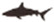	Papuan epaulette shark	Hemiscylliidae	VU	91	0	40	51
40	*Taeniura lymma* 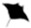	Ribbontail stingray	Dasyatidae	LC	90	0	0	90

**Table 7 animals-14-01761-t007:** First 40 species with their IUCN Red List conservation status on the WatchlistPLUS ranked by the sum of four normalized parameters according to TRACES data from 2014 to 2021: the score of the median number of specimens traded, the score of the linear regression (slope) according to TRACES data, the score of the IUCN Red List conservation status (www.iucnredlist.org (accessed on 23 September 2023)), and the score in vulnerability according to FishBase (http://www.fishbase.org (accessed on 23 September 2023)), resulting in an overall score.

Rank	Species	Common Name	Family	IUCN Red List Status	Overall Score	Median Volume Score	Slope Score	IUCN Score	Vulnerability Score
1	*Chromis viridis*	Blue-green damselfish	Pomacentridae	LC	210	100	100	0	10
2	*Amphiprion ocellaris*	Clown anemonefish	Pomacentridae	LC	157	62	84	0	10
3	*Chelmon rostratus*	Cupperband butterfyfish	Chaetodontidae	LC	67	23	34	0	10
4	*Acanthurus achilles*	Achilles tang	Acanthuridae	LC	68	14	41	0	14
5	*Zebrasoma flavescens*	Yellow tang	Acanthuridae	LC	88	13	18	0	57
6	*Chrysiptera parasema*	Goldtail demoiselle	Pomacentridae	LC	42	9	23	0	10
7	*Chrysiptera cyanea*	Sapphire devil	Pomacentridae	LC	28	6	13	0	10
8	*Nemateleotris magnifica*	Fire goby	Gobiidae	LC	21	5	6	0	10
9	*Centropyge loriculus*	Flame angel	Pomacanthidae	LC	25	5	10	0	10
10	*Zebrasoma velifer*	Sailfin tang	Acanthuridae	LC	59	4	19	0	37
11	*Centropyge acanthops*	Orangeback angelfish	Pomacanthidae	LC	28	4	14	0	10
12	*Acanthurus japonicus*	Japan surgeonfish	Acanthuridae	LC	27	3	13	0	11
13	*Pomacanthus annularis*	Bluering angelfish	Pomacanthidae	LC	44	3	6	0	35
14	*Naso lituratus*	Orangespine unicornfish	Acanthuridae	LC	39	2	3	0	34
15	*Pomacentrus alleni*	Adaman damsel	Pomacentridae	LC	16	2	4	0	10
16	*Pseudochromis fridmani*	Orchid dottyback	Pseudochromidae	LC	15	2	3	0	10
17	*Centropyge tibicen*	Keyhole angelfish	Pomacanthidae	LC	17	2	5	0	10
18	*Centropyge potteri*	Russet angelfish	Pomacanthidae	LC	17	2	5	0	10
19	*Halichoeres chrysus*	Canary wrasse	Labridae	LC	14	2	2	0	10
20	*Dascyllus aruanus*	Whitetail dascyllus	Pomacentridae	LC	31	1	3	0	26
21	*Amphiprion polymnus*	Saddleback clownfish	Pomacentridae	LC	16	1	4	0	10
22	*Dascyllus trimaculatus*	Treespot dascyllus	Pomacentridae	LC	16	1	5	0	10
23	*Ecsenius bicolor*	Bicolor blenny	Blenniidae	LC	13	1	2	0	10
24	*Chaetodon auriga*	Threadfin butterflyfish	Chaetodontidae	LC	32	1	7	0	23
25	*Dascyllus melanurus*	Blacktail humbug	Pomacentridae	LC	15	1	4	0	10
26	*Pomacentrus coelestis*	Neon damselfish	Pomacentridae	LC	13	1	2	0	10
27	*Neosynchiropus ocellatus*	Ocellated dragonet	Callionymidae	N.E.	54	1	2	40	10
28	*Synchiropus picturatus*	Picturesque dragonet	Callionymidae	LC	13	1	2	0	10
29	*Heniochus acuminatus*	Pennant coralfish	Chaetodontidae	LC	20	1	4	0	15
30	*Acanthurus olivaceus*	Orangespot surgeonfish	Acanthuridae	LC	20	1	1	0	18
31	*Amblygobius phalaena*	Whitebarred goby	Gobiidae	LC	13	1	2	0	10
32	*Canthigaster valentini*	Valentin’s sharpnose puffer	Tetraodontidae	LC	12	1	1	0	10
33	*Haemulon flavolineatum*	Frenchg grunt	Haemulidae	LC	44	1	12	0	32
34	*Pterois volitans*	Red lionfish	Scorpaenidae	LC	37	1	2	0	34
35	*Odonus niger*	Red-toothed triggerfish	Balistidae	LC	42	1	1	0	40
36	*Holacanthus ciliaris*	Queen angelfish	Pomacanthidae	LC	38	0	2	0	35
37	*Centropyge flavissima*	Lemonpeel angelfish	Pomacanthidae	LC	12	0	2	0	10
38	*Chrysiptera rollandi*	Rolland’s demoiselle	Pomacentridae	LC	11	0	1	0	10
39	*Amphiprion allardi*	Twobar anemonefish	Pomacentridae	LC	11	0	0	0	10
40	*Synchiropus morrisoni*	Morrison’s dragonet	Callionymidae	LC	11	0	0	0	10

## Data Availability

All data used for the present work are available from public databases (https://comtradeplus.un.org/ (accessed on 23 September 2023) and https://ec.europa.eu/eurostat/data/database (accessed on 23 September 2023)) or can be requested through Regulation (EC) No 1049/2001 of the European Parliament and of the Council of 30 May 2001 regarding public access to European Parliament, Council and Commission documents. TRACES data availability is restricted due to EU guidelines and can be obtained on request: sante-traces@ec.europa.eu.

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
