# Peer review of "An Updated Review of the Marine Ornamental Fish Trade in the European Union"

_animals, 2024, doi:10.3390/ani14121761_

Round 1

Reviewer 1 Report

Comments and Suggestions for Authors

General

Undoubtedly, trade information on marine ornamental fish, derived from a source of market control towards the EU, with unique requirements, a single currency, and a block of countries, allows us to better understand the trade flow of these ornamental fish species. Although implementing TRACES as a system for the entry of ornamental fish into the EU is an advantage, it reduces the sources that can be misled in the analysis. However, in most of these systems, both in the countries of origin of the species (exporters) and those countries that receive them (importers), the scientific denomination of the species (scientific names) is not carried out by experts in the field and generates uncertainty: 1) misidentification of specimens; 2) generalization of names that include specimens of different families/genera/species; 3) non-current identifications (synonymies). The authors do not address this issue in the context of background or discussion. The manuscript could improve by considering the effects on trade flow in ornamental fishes with erroneous names and the tools needed to reduce these complications.

The authors assume that the records' scientific names or species identification are correct. They limit themselves to checking the spelling of the scientific names in the records or even to filling in missing fields. However, they do not have a cross-validation system that allows them to check if indeed the record/identification corresponds. Although they made an effort to filter/clean the database of apparent errors, it is recommendable to raise this point and its implications in the discussion and conclusions of the study.

Specific

1) The Shannon-Wiener Index 220 (H') and Shannon Evenness Index (E') do not support or address the problem. The block of EU countries, except the UK's exit in 2020, would contribute little to the indices in this variable. The block of EU countries does not vary from year to year. The authors should focus on applying these indices to variables related to imported specimens (families, genera, species) rather than on importing countries grouped in the EU block.

2) The WatchlistPLUS differs in the regression analysis given the greater number of years of data. However, the authors warn that given the high variance or absence of data, the significance may be altered or misinterpreted.

3) The document is interesting, based on consolidated information of a single system (TRACES) for ornamental fish trade, destined for a block of countries (EU) with standard regulations, common currency, and common trade provisions. The authors should highlight the contribution of this trade information analysis, especially the difficulties encountered, to propose improvements in the TRACES system, which can be demanded by both exporting and importing countries. Highlight the deficiencies and how to correct the correct identification of the specimens/species/genera/families that are being traded within the EU; through this, we can reach the countries of origin of the specimens and know the condition of exploitation of the natural populations.

Author Response

Undoubtedly, trade information on marine ornamental fish, derived from a source of market control towards the EU, with unique requirements, a single currency, and a block of countries, allows us to better understand the trade flow of these ornamental fish species. Although implementing TRACES as a system for the entry of ornamental fish into the EU is an advantage, it reduces the sources that can be misled in the analysis. However, in most of these systems, both in the countries of origin of the species (exporters) and those countries that receive them (importers), the scientific denomination of the species (scientific names) is not carried out by experts in the field and generates uncertainty: 1) misidentification of specimens; 2) generalization of names that include specimens of different families/genera/species; 3) non-current identifications (synonymies). – The authors thank Reviewer 1 for the constructive criticism provided. The required tools are already in place, as it is a legal obligation of the exporter to use the correct nomenclature on the paperwork needed to legally ship its marine ornamental fishes into the EU. This aspect is even more relevant when using a platform as TRACES, as it was originally developed for veterinary and sanitary control. Providing false information, either by chance or on purpose, will ultimately lead to legal prosecution, with penalties ranging from fees, loss of license to operate or even jail. All fish species have been identified to species-level using WoRMS and synonyms have been changed to the scientific name that is currently valid, from a taxonomic point of view.

The authors do not address this issue in the context of background or discussion. The manuscript could improve by considering the effects on trade flow in ornamental fishes with erroneous names and the tools needed to reduce these complications. - As referred above, it is a legal obligation of the exporter to do so and people who do not obey by this legal framework are committing an illegal action that will lead to prosecution.

The authors assume that the records' scientific names or species identification are correct. They limit themselves to checking the spelling of the scientific names in the records or even to filling in missing fields. However, they do not have a cross-validation system that allows them to check if indeed the record/identification corresponds. – Please see our reply to the previous comment.

Although they made an effort to filter/clean the database of apparent errors, it is recommendable to raise this point and its implications in the discussion and conclusions of the study. - We have assumed that the information is valid (see above), and we have only corrected scientific names and have completed the corresponding family name. The only way to verify if a shipment indeed contains what it is referred in electronic documentation filled by the exporter is to oblige veterinary personal at customs on international airports to inspect each and every single box in a shipment. This is obviously impossible to perform and not reasonable to demand. As such, only by applying strong penalties to those mislabeling the marine ornamental species being traded (e.g., by permanently banning them to ship their fishes to the EU) will it be possible to improve the reliability of the information being provided through TRACES.

Specific

1) The Shannon-Wiener Index 220 (H') and Shannon Evenness Index (E') do not support or address the problem. The block of EU countries, except the UK's exit in 2020, would contribute little to the indices in this variable. The block of EU countries does not vary from year to year. The authors should focus on applying these indices to variables related to imported specimens (families, genera, species) rather than on importing countries grouped in the EU block. - We consider that using these indices provides further insight on this trade. However, if the editor considers that this information is redundant and does not contribute to the goal of our manuscript, the authors will delete it.

2) The WatchlistPLUS differs in the regression analysis given the greater number of years of data. However, the authors warn that given the high variance or absence of data, the significance may be altered or misinterpreted. - For species with high variances or years without any data being reported, we were not able to make a regression analysis and hence these species are NOT featured on the WatchlistPLUS.

3) The document is interesting, based on consolidated information of a single system (TRACES) for ornamental fish trade, destined for a block of countries (EU) with standard regulations, common currency, and common trade provisions. The authors should highlight the contribution of this trade information analysis, especially the difficulties encountered, to propose improvements in the TRACES system, which can be demanded by both exporting and importing countries. - Improvement should be requested by EU member states and it is the obligation of the exporter to report accurate information and the importer to double check that information to make sure it is correct. The exporter knows what she/he is selling, and the importer knows what she/he is buying and paying for (!). The mechanisms to provide reliable information are already in place, with the added value of being dematerialized (no paperwork) and possible to download in a format that can be datamined (ex. xls and cvs files). Continuing to raise awareness that exporters and importers may lose their exporting and importing licenses (respectively) by misspelling the scientific names of the marine ornamental fishes they trade will most likely discourage this practice (especially now that they know that someone is watching closely).

Highlight the deficiencies and how to correct the correct identification of the specimens/species/genera/families that are being traded within the EU; through this, we can reach the countries of origin of the specimens and know the condition of exploitation of the natural populations. - While we may know the level of exploitation, we have no way to know the impact of such exploitation in wild populations, as this can only be reliably assessed by the exporting countries.

Reviewer 2 Report

Comments and Suggestions for Authors

The manuscript ‘An updated review of the marine ornamental fish trade in the European Union’ by Biondo et al. is a comprehensive work that presents strong and exhaustive methods. Other than some minor comments, I find the manuscript a nice contribution to science with a direct application. Of especial relevance are the Conclusions, pointing to future directions that should be followed towards a more sustainable trade of marine ornamental fish.

I provide some specific comments in case these serve to improve the manuscript:

Lines 45-46: any supporting reference?

Line 66: small typo, missing the ‘)’

Line 196: contained, not containing

Line 443: Netherlands, not Netherland

Lines 669-671: this is a copy-paste of the previous recommendation, since it has the same typo…

Line 707: It occurs to me that you could add something regarding potential species misclassification as an additional problem here? And then it would connect with the next paragraph.

Table 1: could you include a total average at the end, not just the 100 proportion? So the average number of total specimens, shipments and value per year imported to the EU is shown.

Table 3: How comes that some countries have no data, i.e.: Slovakia?

Table 4: I understand this is a long table, but some arrangements should be done for it to be better legible, for example not most Family names cut in two.

Author Response

The manuscript ‘An updated review of the marine ornamental fish trade in the European Union’ by Biondo et al. is a comprehensive work that presents strong and exhaustive methods. Other than some minor comments, I find the manuscript a nice contribution to science with a direct application. Of especial relevance are the Conclusions, pointing to future directions that should be followed towards a more sustainable trade of marine ornamental fish. - The authors thank Reviewer 2 for the constructive criticism provided.

I provide some specific comments in case these serve to improve the manuscript:

Lines 45-46: any supporting reference? – Added: Biondo and Burki, 2019, 2020; Biondo and Calado, 2021, Rhyne 2012, 2017; Wabnitz, 2003

Line 66: small typo, missing the ‘)’ – Corrected as suggested

Line 196: contained, not containing - Corrected as suggested

Line 443: Netherlands, not Netherland - Corrected as suggested

Lines 669-671: this is a copy-paste of the previous recommendation, since it has the same typo… - corrected typo and re-phrased.

Line 707: It occurs to me that you could add something regarding potential species misclassification as an additional problem here? And then it would connect with the next paragraph. – In order to best accommodate the recommendation by reviewer 2 we added the following sentence in our revised manuscript: “If up-to-date information is made available on the conservation status of the most traded marine ornamental fish species, watchlists may allow policy makers to take action before reaching tipping points and fisheries need to be closed” (please refer to lines 722 - 725).

Table 1: could you include a total average at the end, not just the 100 proportion? So the average number of total specimens, shipments and value per year imported to the EU is shown. – Corrected as suggested.

Table 3: How comes that some countries have no data, i.e.: Slovakia? – It may have been a sporadic import by a single company: From our data it appears that this country only directly imported 61 fishes in 2015 from third countries (for example Indonesia). However, it is possible that Slovakia may have imported more fishes from other EU countries, as this internal trade (intra EU) will not be featured in the present study.

Table 4: I understand this is a long table, but some arrangements should be done for it to be better legible, for example not most Family names cut in two. –  Corrected as suggested.

Reviewer 3 Report

Comments and Suggestions for Authors

I have made edits and comments directly into the manuscript and provided a summary of points at the end of the manuscript for the authors' to consider for improving the manuscript.  The paper is important given the current interest by protagonists to the CITES Secretariat over sustainability concerns in the aquarium trade, especially for marine ornamental fish.   

Comments on the Quality of English Language

The writing is fine, I have made some edits for the authors' to consider.

Author Response

I have made edits and comments directly into the manuscript and provided a summary of points at the end of the manuscript for the authors' to consider for improving the manuscript. The paper is important given the current interest by protagonists to the CITES Secretariat over sustainability concerns in the aquarium trade, especially for marine ornamental fish. - The authors thank reviewer 3 for the constructive criticism and we have tried to best accommodate the comments and suggestions provided by correcting the manuscript accordingly and answering each of the comments provided (please refer to the annotated manuscript with Track Changes). We would like to highlight that our work only addresses the EU market in terms of number of specimens and species traded. Although the EU is certainly one of the most relevant players in the marine aquarium industry, it would be speculative (and most likely erroneous) to extrapolate trends for other import markets. To date, it is impossible to reliably compare the trends reported here for the EU with those referring to the USA, the Peoples Republic of China and any other relevant markets for marine ornamental fishes. Disentangling the effects of Covid-19 and Brexit on the EU market may only be possible in the next five years or so. This does not mean that data and potential trends, such as the ones here reported, should not be made available for this, or any other commercial activity, namely, to raise awareness and urge decision makers to continue to gather reliable data so informed decisions can be made (based on facts and not opinions).

Reviewer 4 Report

Comments and Suggestions for Authors

The manuscript is well-structured, clearly written, and provides valuable insights into the marine ornamental fish trade. The methodology is robust, and the results are presented effectively. The discussion and conclusions are well-founded, offering practical recommendations for improving the monitoring and management of this trade.

Minor comments

  • Ensure that all acronyms (e.g., IUCN) are clearly defined when first mentioned.
  • Consistently use terms throughout the manuscript to avoid any confusion.
  • The quality of figures can be improved, such as Axis titles, and legends.
  • Highlight potential areas for future research, particularly in assessing the long-term impacts of the proposed TRACES enhancements and the effectiveness of the watchlist in mitigating overexploitation risks.
  • Discuss any limitations of the current study and suggest how future studies could address these gaps.
  • The MS is more descriptive rather than analytic, the authors can do some statistical analysis.
  • Include a brief discussion on the potential impact of recent global events, such as Brexit and the COVID-19 pandemic, on the trade and how the proposed recommendations could address these challenges.

The manuscript is a valuable contribution to the field of marine ornamental fish trade and conservation. With the minor corrections addressed, it will provide even clearer and more impactful insights into the trade and its management.

Author Response

The manuscript is well-structured, clearly written, and provides valuable insights into the marine ornamental fish trade. The methodology is robust, and the results are presented effectively. The discussion and conclusions are well-founded, offering practical recommendations for improving the monitoring and management of this trade. - The authors thank Reviewer 4 for the constructive criticism provided.

Minor comments

  • Ensure that all acronyms (e.g., IUCN) are clearly defined when first mentioned. – Corrected as suggested.
  • Consistently use terms throughout the manuscript to avoid any confusion. – Corrected as suggested.
  • The quality of figures can be improved, such as Axis titles, and legends. – Corrected as suggested.
  • Highlight potential areas for future research, particularly in assessing the long-term impacts of the proposed TRACES enhancements and the effectiveness of the watchlist in mitigating overexploitation risks. – It is the authors opinion that we already address this topic in our conclusions (please refer to lines 573 – 539, 710 – 725, 731 – 736, 778 - 785). However, if either Reviewer 4 or the Editor can provide a clearer guidance on how we can reinforce our “take home message”, the authors will certainly add that information to the manuscript.
  • Discuss any limitations of the current study and suggest how future studies could address these gaps. – In our manuscript we consider having addressed the limitations of our findings. As more years of data become available, the WatchlistPLUS can be made more reliable; the four parameters of the Watchlist are weight the same.
  • The MS is more descriptive rather than analytic, the authors can do some statistical analysis. – in addition to our regression analysis and our diversity indexes (please refer to our supplementary material), we are open to include new statistical analysis if either Reviewer 4 or the Editor can kindly indicate which ones will better suit our goals.
  • Include a brief discussion on the potential impact of recent global events, such as Brexit and the COVID-19 pandemic, on the trade and how the proposed recommendations could address these challenges. please see comments to reviewer 3 and lines 583 – 589, 601 - 602, 633 – 636, 779 - 782.

The manuscript is a valuable contribution to the field of marine ornamental fish trade and conservation. With the minor corrections addressed, it will provide even clearer and more impactful insights into the trade and its management.

Round 2

Reviewer 3 Report

Comments and Suggestions for Authors

The authors have done well to address concerns with the original draft manuscript.  I appreciate the Authors are promoting TRACES and it may be the logical data collection point for ornamental imports to the EU, but there is still a need need for further research to consider economic and global factors in assessing trade effects on species populations, highlighting the importance of a more holistic approach to understanding the trade's impacts.  I have no further comments.